# A prospective model of the potential clinical and economic impact of cervical cancer screening supported by a mobile phone app

**Fredrick Chite Asirwa**[1], **Brian W. Bresnahan**[2,3], **Faith Yego**[4], **Dana Duncan**[5], **James K. Karichu**[6], **Louis P. Garrison Jr.**[3,7]*

**1** International Cancer Institute, Eldoret, Kenya, **2** Department of Radiology, School of Medicine, University of Washington, Seattle, Washington, United States of America, **3** The Comparative Health Outcomes, Policy, and Economics (CHOICE) Institute, University of Washington, Seattle, Washington, United States of America, **4** Department of Health Policy, Management, and Human Nutrition, Moi University School of Public Health, Eldoret, Kenya, **5** Roche Information Solutions, Roche Diagnostics, Santa Clara, California, United States of America, **6** Global Access & Policy, Roche Diagnostics Solutions, Inc., Pleasanton, California, United States of America, **7** VeriTech Corporation, Mercer Island, Washington, United States of America

* lgarrisn@uw.edu

## Abstract

### Introduction

Cervical cancer is a preventable and highly curable disease when detected early and adequately treated, yet it remains the leading cause of cancer-related death in women in Kenya due to low screening coverage and treatment. Implementing World Health Organization screening guidelines for human papillomavirus (HPV) is challenging due to the complex logistics of result return and follow-up requiring multiple clinic visits. Increasing the use of mobile technology can support follow-up care in cervical cancer screening programs.

### Methods

We developed a prospective clinico-economic model to assess the potential impact of a mobile phone-based application ("app") communicating laboratory results and recommendations to improve follow-up care for cervical cancer screening in Kenya. The model is structured to simulate a three-visit pathway for HPV-based screening used in a clinical trial of the app and based on epidemiological data, clinical guideline-based workflow, and patient-based behavioral pathways. Published literature, expert elicitation, and time-and-motion observations were used to estimate clinical data, care pathways, and visit-related costs. This analysis was conducted from a base-case healthcare system perspective with a scenario from a "limited" societal perspective.

### Results

In a simulated cohort of women using the app-based intervention compared to conventional care, with 10,000 women in each arm, use of the app is projected to increase healthcare costs by $12.53 per enrolled woman during the trial period and to detect and treat an additional 247 women—229 with precancerous cervical lesions and 18 with

**Data availability statement:** All relevant data needed to replicate the modeling results in this paper are in the Table 1 (probabilities) and Table 2 (costs).

**Funding:** Roche Diagnostics funded the economic analysis through a contract with Veritech Corporation, providing support for LPG, BWB, and FY.

**Competing interests:** We have read the journal's policy on competing interests, and the authors of this manuscript have the following competing interests: FCA, BWB, LPG, FY received research support from Roche Diagnostics. DD and JKK are employees of Roche Diagnostics.

cervical cancer. The incremental cost-effectiveness ratio of the app versus conventional care was $174 per case detected and treated. This would be cost-saving given the average lifetime cost per cervical cancer case of $1,000–$3,000.

## Conclusion

Use of a mobile phone-based app is costlier than conventional screening but by improving visit compliance, it can be a cost-effective and cost-saving strategy to enhance detection and treatment in cervical cancer screening programs.

## Introduction

Cervical cancer is the fourth most common cause of cancer among women worldwide and has wide geographical variation in incidence and mortality. Over 90% of cervical cancer deaths occur in low- and middle-income countries [1]. Measures for eliminating cervical cancer exist but are not widely implemented in regions of the world where the disease burden is highest. In 2020, the World Health Organization (WHO) set a 2030 target to screen 70% of women with a high-performance cervical cancer test by 35 and 45 years of age and to treat 90% of women identified with cervical precancer or cancer [2,3]. Achieving these targets is challenging for many countries with a high disease burden due to the multiple clinic visits required to complete screening and treatment. Alternative screening approaches are being evaluated—with and without DNA testing [4]. In any approach, effective systems for increasing patient compliance with follow-up care are critical to achieving the WHO goal of cervical cancer elimination.

Mobile phone technology holds immense potential to increase visit compliance. With mobile phone coverage of around 90%, most women in Kenya may benefit from a mobile-phone application (app)-based intervention to increase the effectiveness of cervical cancer screening programs [5,6]. Mobile phone telecommunication using text messaging has been used effectively in cervical cancer screening programs to provide reminders for follow-up care, education, and electronic specimen tracking [7,8]. Mobile app-based interventions offer the advantage over text messages of tailoring content to match the needs and preferences of users and provide multimedia content to enhance participation and further motivate behavior change [9–11]. When strategically integrated into cervical cancer screening programs, mobile apps may increase completion rates for screening and treatment.

The clinical impact and cost-effectiveness of using innovative apps to increase visit compliance in cervical cancer screening programs in low-and middle-income countries (LMICs) is largely unknown. However, there has been important research on developing and implementing mobile apps for cervical cancer screening in Kenya, in particular [12–15]. We modeled the potential clinical and economic value of a mobile app to boost adherence and treatment by women enrolled in a cervical cancer screening program relative to the estimated cost. This model can be used to evaluate the potential clinical and economic impact of an app used for cervical cancer screening in Western Kenya. The results of this analysis can support policy and program implementation discussions for using mobile app technology in cervical cancer screening programs in Kenya and other high-burden settings.

## Methods

### Overview

We developed a prospective clinico-economic model alongside an ongoing randomized-controlled trial (RCT) to assess the potential economic impact on cervical cancer screening

in clinics in western and eastern Kenya supported by an innovative mobile phone-based app used to communicate care outcomes and recommendations to patients. Our model was based on the clinical protocol used in an RCT study (ITH-HPV-548; RD006031) assessing the clinical consequences associated with using a mobile app compared to standard of care to aid communication to women about human papillomavirus (HPV) testing results, follow-up care recommendations, and cervical cancer screening outcomes. The timeframe for this prospective model included a basic three-visit schedule (usually completed in less than six months), aligned with guideline-based care. The model is structured to simulate expected pathways in Kenya for HPV-based cervical cancer screening and subsequent care based on biologic information (epidemiology), clinical guideline-based workflow pathways, and patient-based behavioral pathways. These alternative pathways influence the proportion of patients experiencing different clinical events in a simulated flow for a cohort of patients using a cost-consequences framework. Key consequences include intermediate endpoints (such as test results) and final endpoints (viz., pre-cancerous and cancer cases detected and treated). The best available relevant evidence influenced our parameter values for base-case values and ranges for sensitivity analyses.

Multiple sources of information were used, including data from: (a) published literature and publicly-available websites reporting on cervical cancer screening, treatments, costs, and published guidelines, (b) expert elicitation, primarily from "subject matter experts" (SMEs) representing a not-for-profit private cancer center in Western Kenya conducting an HPV-screening RCT, as well as experts in other regions of Kenya; (c) a time-and-motion study capturing time and resources involved in standard cervical cancer screening visits and follow-up care (private clinic and public hospital settings); and (d) testing-related and procedure-related cost estimates collected through key-informant interviews (multiple observations). The software acquisition cost of the mobile app was not included in the model.

This prospective clinico-economic analysis was conducted from two perspectives—a base-case healthcare system perspective and a scenario considering a "limited" societal perspective [16]. The latter includes estimates of patient opportunity costs associated with expected incremental differences in reduced time required to attend clinic visits for the intervention and control groups. We begin with a cost-consequences approach since there are multiple intermediate and final outcomes being generated from the RCT and projected real-world pathways, such as differences in patients receiving post-screening follow-up care, including: recommended visit attendance, receiving colposcopy and/or biopsy for higher-risk cases, and receiving guideline-consistent pre-cancerous lesions and cancer-related treatment. The consequences for patients and expected patient flow processes result in patients following distinct care pathways subsequent to HPV-based screening result availability. Our model structure also allows for the calculation of some specific cost-per-consequence metrics (i.e., cost-effectiveness measures such as cost-per-case detected and cost-per-cancer-case avoided).

These between-group differences—those using the mobile app or not—are expected to be influenced by differential follow-up visit compliance rates, which is due presumably to the improved ability to notify and communicate with patients using the mobile app. Our prior expectation was that there are likely to be incremental cost-savings associated with the intervention group due to a reduced number of clinical follow-up visits required to receive "normal" results, i.e., negative high-risk human papillomavirus (hrHPV) test results. In addition, given the improved communication opportunities with the mobile app, we expect a higher prevalence of attendance for recommended, "clinically necessary" follow-up visits. Due to better follow-up visit compliance, we also expect higher total costs for patients with the mobile app for visits associated with performing colposcopies and biopsies to reveal cervical intraepithelial neoplasia (CIN) status or cancer diagnosis and to make pre-cancer and cancer-related

treatment decisions. Likewise, we expect higher eventual-treatment costs consistent with clinical guideline-based care in the intervention group, due to the relative ease of patient contact via the mobile application. Our model allowed for a per-patient cost for navigation or phone follow-up for each group.

We developed a decision-tree model structure mirroring the guideline-based clinical practices and generally expected patient and decision-making flow in the RCT for obtaining and disseminating cervical cancer screening consequences and recommendations. We constructed an Excel-based model calculating expected incremental costs and consequences within and between the randomized groups. To inform model structure and to parameterize the model, we conducted a targeted assessment of the cost-related and cost-effectiveness-related literature, as well as select clinical and epidemiologic literature and clinical agency websites. Our aim was to identify the prevalence of cervical cancer, clinical outcomes, and costs for cervical cancer screening, diagnosis, and treatment. For the literature assessment, the following databases were searched (PubMed, OVID, Embase, ProQuest, and Google Scholar). The search strategy applied to all databases used the following keywords ("cost-effectiveness", "cervical cancer," and "Kenya"). We also used specific keywords such as "cervical cancer screening in Kenya;" "mobile applications and cervical cancer in Kenya;" "economics of cervical cancer screening in Kenya;" "consequences of cervical cancer in Kenya;" and "scaling up cervical cancer in Kenya." We focused on literature specific to Kenya from 2010–2023. We utilized only articles that were in English. All abstracts meeting the search criteria were assessed and reviewed. Full-text reviews were conducted for publications or reports deemed relevant to informing model parameterization.

We also relied on the informed judgment of the study team to finalize the model parameters with assumptions based on the literature and on interviews with site staff. Thus, we were able to define clinical pathways and a clinical-outcomes structure, and to generate a range of base-case estimates derived from relevant probabilities and costs. In addition, the literature-based parameters were supplemented with estimates by SMEs in Kenya. We (see supplemental Inclusivity in Global Research Questionnaire) collaborated with clinical and clinic administration professionals and study investigators to refine estimates for the model clinical parameters, specify ranges of parameter estimates, and develop plausible cost estimates for services and interventions. Ethics approval for the expert elicitation was obtained from KEMRI (Protocol #4217), and verbal informed consent was obtained from participants prior to conducting the interviews (between May 15, 2023, and August 15, 2023). For the verbal consent process, the study was explained to the participants who were selected based on their expertise in the study topic, and once they agreed to participate, the consent was documented and the interview conducted. All informed consenting processes were approved by the IRB.

Our primary base-case estimates are structured to first report healthcare process metrics, such as the number of patients screened, the number of visits attended, and counts of patients receiving various interventions (colposcopies, biopsies, pre-cancer, and cancer treatment), accounting for preliminary loss-to-follow-up (LTFU) assumptions. We also report group-related aggregate cost estimates for each treatment arm, associated with the use of these services and visits, as well as per-patient cost estimates. Since the time horizon for the primary analysis is short-term (i.e., less than one year to complete the 3 visits), there is no strong need to perform discounting of costs or outcomes. Given the high likelihood of patients not returning for follow-up visits, we specifically report statistics related to visit compliance. Lastly, we conducted selected one-way sensitivity analyses—altering one key model parameter at a time—to provide information about the expected impact on results due to changing different variables related to probabilities, costs, or follow-up rates. Additional information describing the three-visit study protocol and what tests or interventions were administered to patients at specific visits is provided in the Supporting information section.

## Results

Our prospective clinico-economic model structure aligned with an RCT based on standard clinical practice in Kenya. Fig 1 presents the trial clinical pathways prescribed in the protocol.

Table 1 presents model input parameters for probabilities, including the expected likelihood of attending recommended follow-up visits. On the first visit, visual inspection with acetic acid (VIA) or visual inspection with Lugol's iodine (VILI) and HPV sample collection for testing are performed. The probabilities provide estimates for HPV test findings, for the likelihood of receiving follow-up care at a second healthcare encounter, and for receiving pre-cancer treatment at a third encounter as needed. Based on the literature and expert clinical judgment, the vast majority (80%) of patients are expected to be VIA- and HPV- and thus have no clinical reason to physically attend a second healthcare visit. The remainder of the patients are categorized into four groups with their expected percentages: VIA+ & hrHPV- (& no tumor) (3%), VIA-/VIA+ and/or hrHPV+ (non 16/18 & no tumor) (6%), VIA-/VIA+ and/or hrHPV+ (16/18 & no tumor) (9%), and suspicious for CA (VIA+ with Gross Tumor) (2%).

Cost parameters used in the model are presented in Table 2, in estimated 2022 U.S. dollars. The costs for routine screening visits are included ($28.5), as well as the average cost of colposcopies ($35) and biopsies ($105) if subsequent diagnostic tests are deemed clinically necessary and performed after the HPV results are obtained. In addition, costs for the time of the clinical staff to support downloading the app at the screening visit are included ($3), making a screening visit $31.5 for the app arm. Average costs related to contacting patients if they do not attend a visit are included for completeness ($2 for the control arm and $0 for the app arm). The estimates for pre-cancer treatment were $348 per episode (generally cryotherapy or loop electrosurgical excision procedure (LEEP). Cancer treatment per patient was estimated to be $425 which currently only includes the additional cost of local lesion therapy.

Table 3 summarizes the visit and cost results of the model projection. In a simulated cohort of 10,000 patients in each arm, it is projected that the app-based intervention

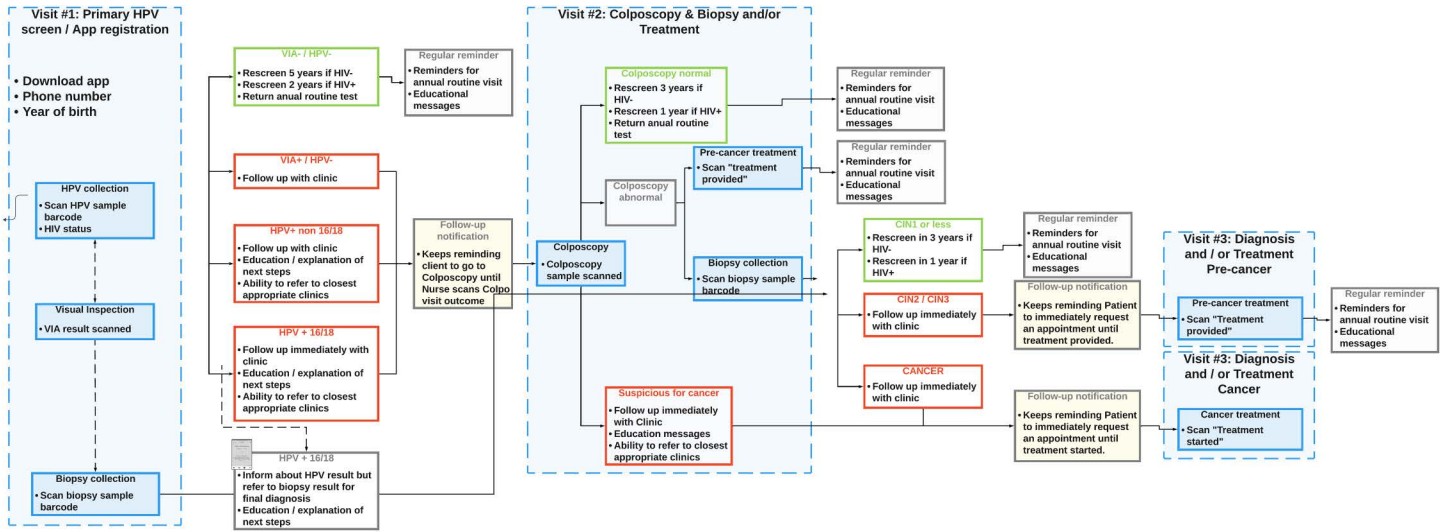

**Fig 1. Clinical trial workflow protocol.**

**Table 1. Probabilities of clinical events.**

| | Likely | Low | High | Source |
|---|---|---|---|---|
| **Visit #1: Primary HPV/Screen App Registration** | | | | |
| *Initial Screen Result* | | | | |
| 1. VIA- & HPV- | 0.8 | 0.64 | 0.96 | [17] |
| 2. VIA+ & hrHPV- (& no tumor) | 0.03 | 0.024 | 0.036 | [18] |
| 3. VIA-/VIA+ and/or hrHPV+ (non 16/18 & no tumor) | 0.06 | 0.048 | 0.072 | [19] |
| 4. VIA-/VIA+ and/or hrHPV+ (16/18 & no tumor) | 0.09 | 0.072 | 0.108 | [20,21] |
| 5. Suspicious for CA (VIA+ with Gross Tumor) | 0.02 | 0.016 | 0.024 | Assumption—Expert Judgment/Elicitation |
| **Conditional Probability of Return for Second Visit or Follow-up Visit** | | | | |
| *Control Arm (Return to find out HPV results)* | | | | |
| 1. VIA- & HPV- | 0.6 | 0.48 | 0.72 | [17] |
| 2. VIA+ & hrHPV- (& no tumor) | 0.6 | 0.48 | 0.72 | [22] |
| 3. VIA-/VIA+ and/or hrHPV+ (non 16/18 & no tumor) | 0.6 | 0.48 | 0.72 | [19] |
| 4. VIA-/VIA+ and/or hrHPV+ (16/18 & no tumor) | 0.6 | 0.48 | 0.72 | [20,21] |
| 5. Suspicious for CA (VIA+ with Gross Tumor) | 0.75 | 0.6 | 0.9 | Assumption—Expert Judgment/Elicitation |
| *Mobile App Arm (Informed by App; If return, for Colposcopy)* | | | | |
| 1. VIA- & HPV- | 0 | 0 | 0 | [17] |
| 2. VIA+ & hrHPV- (& no tumor) | 0.9 | 0.72 | 0.95 | [22] |
| 3. VIA-/VIA+ and/or hrHPV+ (non 16/18 & no tumor) | 0.9 | 0.72 | 0.95 | [19] |
| 4. VIA-/VIA+ and/or hrHPV+ (16/18 & no tumor) | 0.9 | 0.72 | 0.95 | [20,21] |
| 5. [Suspicious for CA (VIA+ with Gross Tumor) [See below]] | 0.85 | 0.68 | 0.95 | Assumption—Expert Judgment/Elicitation |
| **Visit #2: Colposcopy & Biopsy and/or Treatment** | | | | |
| *Visit Result (Conditional on Arm)* | | | | |
| *Subpop 1. VIA- & HPV-* | | | | |
| A. Colposcopy Normal | 0.28 | 0.224 | 0.336 | [23] |
| B. Colposcopy Abnormal—Pre-cancer Treatment (As % of All) | 0.32 | 0.256 | 0.384 | [17] |
| C. Colposcopy Abnormal—Biopsy Collection CIN1 or less (As % of All) | 0.33 | 0.264 | 0.396 | [24] |
| D. Colposcopy Abnormal—Biopsy Collection CIN2/CIN3 (As % of All) | 0.06 | 0.048 | 0.072 | [24] |
| E. Colposcopy Abnormal—Suspicious for CA | 0.01 | 0.008 | 0.012 | Per protocol |
| *Subpop 2. VIA+ & hrHPV- (& no tumor)* | | | | |
| A. Colposcopy Normal | 0.28 | 0.224 | 0.336 | [23] |
| B. Colposcopy Abnormal—Pre-cancer Treatment (As % of All) | 0.32 | 0.256 | 0.384 | [17] |
| C. Colposcopy Abnormal—Biopsy Collection CIN1 or less (As % of All) | 0.33 | 0.264 | 0.396 | [24] |
| D. Colposcopy Abnormal—Biopsy Collection CIN2/CIN3 (As % of All) | 0.06 | 0.048 | 0.072 | [24] |
| E. Colposcopy Abnormal—Suspicious for CA | 0.01 | 0.008 | 0.012 | Per protocol |
| *Subpop 3. VIA-/VIA+ and/or hrHPV+ (non 16/18 & no tumor)* | | | | |
| A. Colposcopy Normal | 0.28 | 0.224 | 0.336 | [23] |
| B. Colposcopy Abnormal—Pre-cancer Treatment (As % of All) | 0.32 | 0.256 | 0.384 | [17] |
| C. Colposcopy Abnormal—Biopsy Collection CIN1 or less (As % of All) | 0.33 | 0.264 | 0.396 | [24] |
| D. Colposcopy Abnormal—Biopsy Collection CIN2/CIN3 (As % of All) | 0.06 | 0.048 | 0.072 | [24] |
| E. Colposcopy Abnormal—Suspicious for CA | 0.01 | 0.008 | 0.012 | Per protocol |
| *Subpop 4. VIA-/VIA+ and/or hrHPV+ (16/18 & no tumor)* | | | | |
| A. Colposcopy Normal | 0.28 | 0.224 | 0.336 | [23] |
| B. Colposcopy Abnormal—Pre-cancer Treatment (As % of All) | 0.32 | 0.256 | 0.384 | [17] |
| C. Colposcopy Abnormal—Biopsy Collection CIN1 or less (As % of All) | 0.33 | 0.264 | 0.396 | [24] |
| D. Colposcopy Abnormal—Biopsy Collection CIN2/CIN3 (As % of All) | 0.06 | 0.048 | 0.072 | [24] |
| E. Colposcopy Abnormal—Suspicious for CA (As % of All) | 0.01 | 0.008 | 0.012 | Per protocol |

*(Continued)*

**Table 1.** (Continued)

| | Likely | Low | High | Source |
|---|---|---|---|---|
| *Subpop 5. Suspicious for CA (VIA+ with Gross Tumor)* | | | | |
| F.  First visit suspicious of CA—Biopsy CIN1 or less | 0.1 | 0.08 | 0.12 | Assumption—Expert Judgment/Elicitation |
| G.  First visit suspicious of CA—Biopsy CIN2/CIN3 | 0.4 | 0.32 | 0.48 | Assumption—Expert Judgment/Elicitation |
| H. First visit suspicious of CA—Cancer | 0.5 | 0.4 | 0.6 | Assumption—Expert Judgment/Elicitation |
| **Visit #3: Diagnosis and/or Treatment (Pre-cancer or Cancer)** | | | | |
| *Conditional Probability (on Visit #2 Outcome) of RETURN for Visit 3* | | | | |
| **Control** | | | | |
| A.  Colposcopy Normal | 0 | 0 | 0 | Assumption—Expert Judgment/Elicitation |
| B.  Colposcopy Abnormal—Pre-cancer Treatment (As % of All) | 0.5 | 0.4 | 0.575 | Assumption—Expert Judgment/Elicitation |
| C.  Colposcopy Abnormal—Biopsy Collection CIN1 or less (As % of All) | 0.5 | 0.4 | 0.575 | Assumption—Expert Judgment/Elicitation |
| D.  Colposcopy Abnormal—Biopsy Collection CIN2/CIN3 (As % of All) | 0.5 | 0.4 | 0.575 | Assumption—Expert Judgment/Elicitation |
| E.  Colposcopy Abnormal—Suspicious for CA (As % of All) | 0.8 | 0.64 | 0.88 | Assumption—Expert Judgment/Elicitation |
| *Conditional Probability (on Visit #1 Outcome and of RETURN for Planning Visit 2)* | | | | |
| F.   First visit suspicious of CA—Biopsy CIN1 or less | 0.8 | 0.64 | 0.96 | Assumption—Expert Judgment/Elicitation |
| G.  First visit suspicious of CA—Biopsy CIN2/CIN3 | 0.8 | 0.64 | 0.96 | Assumption—Expert Judgment/Elicitation |
| H. First visit suspicious of CA—Cancer | 0.9 | 0.72 | 0.99 | Assumption—Expert Judgment/Elicitation |
| **Mobile App** | | | | |
| A.  Colposcopy Normal | 0 | 0 | 0 | Assumption—Expert Judgment/Elicitation |
| B.  Colposcopy Abnormal—Pre-cancer Treatment (As % of All) | 0 | 0 | 0 | Assumption—Expert Judgment/Elicitation |
| C.  Colposcopy Abnormal—Biopsy Collection CIN1 or less (As % of All) | 0 | 0 | 0 | Assumption—Expert Judgment/Elicitation |
| D.  Colposcopy Abnormal—Biopsy Collection CIN2/CIN3 (As % of All) | 0.8 | 0.64 | 0.88 | Assumption—Expert Judgment/Elicitation |
| E.  Colposcopy Abnormal—Suspicious for CA (As % of All) | 0.9 | 0.72 | 0.99 | Assumption—Expert Judgment/Elicitation |
| *Conditional Probability (on Visit #1 Outcome and of RETURN for Planning Visit 2)* | | | | |
| F.   First visit suspicious of CA—Biopsy CIN1 or less | 0 | 0 | 0 | Assumption—Expert Judgment/Elicitation |
| G.  First visit suspicious of CA—Biopsy CIN2/CIN3 | 0.9 | 0.72 | 0.99 | Assumption—Expert Judgment/Elicitation |
| H. First visit suspicious of CA—Cancer | 0.9 | 0.72 | 0.99 | Assumption—Expert Judgment/Elicitation |
| **Treatment (Conditional on Visit #2 or Follow-up Visit Outcome)** | | | | |
| **Colposcopy Normal** | | | | |
| Pre-cancer Treatment | 0 | 0 | 0 | Protocol |
| Cancer Treatment | 0 | 0 | 0 | Protocol |
| Colposcopy Abnormal—Pre-cancer Treatment | | | | |
| Pre-cancer Treatment | 1 | 1 | 1 | Protocol |
| Colposcopy Abnormal—Biopsy Collection CIN1 or less | | | | |
| Pre-cancer Treatment | 0 | 0 | 0 | Protocol |
| Colposcopy Abnormal—Biopsy Collection CIN2/CIN3 | | | | |
| Pre-cancer Treatment | 1 | 1 | 1 | Protocol |
| Colposcopy Abnormal—Biopsy Collection—Cancer | | | | |
| Cancer Treatment | 1 | 1 | 1 | Protocol |
| **Control** | | | | |
| Screening visit suspicious of CA—Biopsy CIN1 or less | | | | |
| Pre-cancer Treatment | 0 | 0 | 0 | Protocol |
| Screening visit suspicious of CA—Biopsy CIN2/CIN3 | | | | |
| Pre-cancer Treatment | 1 | 1 | 1 | Protocol |
| Screening visit suspicious of CA—Cancer | | | | |
| Cancer Treatment | 1 | 1 | 1 | Protocol |

*(Continued)*

**Table 1.** (Continued)

| | Likely | Low | High | Source |
|---|---|---|---|---|
| **Mobile App** | | | | |
| Screening visit suspicious of CA—Biopsy CIN1 or less | | | | |
| Pre-cancer Treatment | 0 | 0 | 0 | Protocol |
| Screening visit suspicious of CA—Biopsy CIN2/CIN3 | | | | |
| Pre-cancer Treatment | 1 | 1 | 1 | Protocol |
| Screening visit suspicious of CA—Cancer | | | | |
| Cancer Treatment | 1 | 1 | 1 | Protocol |

Source: Authors' estimates.

**Table 2.** Input parameters—Costs.

| *Healthcare System Perspective* | | | | |
|---|---|---|---|---|
| **Cost per Visit:** | **Likely** | **Low** | **High** | **Source** |
| *Visit #1: Primary HPV/Screen App Registration* | | | | |
| Control Arm | $28.50 | $22.80 | $34.20 | [25]; Author calculations |
| Mobile App Arm | $31.50 | $25.20 | $37.80 | HPV test [25,26]; plus time costs |
| *Post-Visit 1: Navigation/Notification* | | | | |
| Control Arm | $2.00 | $1.60 | $2.40 | Assumption—Expert Judgment/Elicitation |
| Mobile App Arm | $0.00 | $0.00 | $0.00 | Assumption—Expert Judgment/Elicitation |
| *Visit #2: Colposcopy & Biopsy and/or Treatment* | | | | |
| Cost of Information-Only Visit | $5.00 | $4.00 | $6.00 | Assumption—Expert Judgment/Elicitation |
| Base Encounter Cost | $28.50 | $22.80 | $34.20 | Assumption—Expert Judgment/Elicitation |
| Colposcopy (No biopsy) | $35.00 | $28.00 | $42.00 | Assumption—Expert Judgment/Elicitation |
| Biopsy Incremental Cost | $105.00 | $84.00 | $126.00 | Assumption—Expert Judgment/Elicitation |
| Post-Visit 2: Navigation/Notification | | | | |
| Control Arm | $2.00 | $1.60 | $2.40 | Assumption—Expert Judgment/Elicitation |
| Mobile App Arm | $0.00 | $0.00 | $0.00 | Assumption—Expert Judgment/Elicitation |
| Visit for Treatment Initiation: Diagnosis and/or Treatment (Pre-cancer or Cancer) | | | | |
| Follow-up on Visit 1 Biopsy for Treatment Planning | $15.00 | $12.00 | $18.00 | Assumption—Expert Judgment/Elicitation |
| Pre-cancer Treatment | $348.00 | $278.40 | $417.60 | Assumption—Expert Judgment/Elicitation |
| Cancer Treatment | $425.00 | $40.00 | $510.00 | Assumption—Expert Judgment/Elicitation |
| *Societal Perspective* | | | | |
| Patient Travel Cost and Time Cost | | | | |
| All visits—Mean patient time cost (missed work) | 3 | 2.4 | 3.6 | https://www.fke-kenya.org/Minimumwages |
| All visits—Mean Total Travel Cost per Visit (round trip) | 3 | 2.4 | 3.6 | https://www.fke-kenya.org/Minimumwages |
| Total Societal Cost per Visit | 6 | 4.8 | 7.2 | Calculated |
| All visits—Mean select out-of-pocket costs (childcare and meals) | 2 | 1.6 | 2.4 | [27] (semi-integrated visit); Author calculations |

Source: Authors' estimates.

would generate increased healthcare costs of $125,278 (or $12.53 per enrolled patient) over the three-visit study period. Based on an epidemiological assumption, 80% of screened patients are VIA- and HPV-, and follow-up is not required. Unlike the intervention group, patients in the control arm need to return for a second visit to find out

**Table 3. Summary of key cost impacts.**

| | Control Arm | | Mobile App | | Difference |
|---|---|---|---|---|---|
| | **Number** | **Total** | **Number** | **Total** | |
| First visit—Screening and App Loading Costs | 10,000 | $285,000 | 10,000 | $315,000 | $30,000 |
| Mobile phone notification of controls post-Visit 1 | 10,000 | $20,000 | 0 | $0.00 | $−20000 |
| Second Visit—Non-suspicious with Colposcopy | 1,080 | $68,580 | 1,620 | $102,870 | $34,290 |
| Second Visit—Non-suspicious without Colposcopy | 4,800 | $24,000 | 0 | $0.00 | $−24000 |
| Biopsies | 632 | $66,360 | 848 | $89,040 | $22,680 |
| Mobile call notification of controls post-Visit 2 | 632 | $1,264 | 0 | $0.00 | $−1264 |
| Return for V2 Treatment Initiation for V1 Ca Suspicious | 132 | $1,973 | 149 | $228 | $255 |
| Return for V3 CIN Status (CIN1) | 1 | $5.00 | 0 | $0.00 | $−4.50 |
| Pre-Cancer Treatments (Visit 3) | 444 | $154,442 | 673 | $234,211 | $79,769 |
| Cancer Treatments (V3) | 70 | $29,835 | 84 | $35,611 | $5,776 |
| *Tota Detection Costs* | – | *$463,940* | – | *$506,910* | *$42,970* |
| *Trial Period Total Detection and Treatment Costs* | – | *$651,454* | – | *$776,732* | *$125,278* |

Source: Authors' estimates.

their status. This generates many visits for the VIA-/HPV- patients: we assume the cost is $5 for on-site notification alone vs. $64 for a full return visit with colposcopy. Factoring in per-protocol colposcopy for the patients VIA+ or hrHPV+, overall, this results in an additional 4,257-second visits in the control arm, but the net effect is that the intervention arm costs $6,206 more. The increased cost due to the intervention is projected, however, to produce potential health gains in the cohort in that, as shown in Table 4, an additional 229 patients who are at high risk of cancer are detected and receive pre-cancer treatment. Furthermore, 18 more patients are detected as having cancer and will receive treatment.

From a cost-effectiveness perspective, we can calculate (Table 5) the incremental detection cost up to the point of either pre-cancer treatment or cancer treatment. Thus, excluding the projected treatment costs from the total, the mobile app arm generates $42,970 in additional costs to detect and treat an additional 247 patients. The incremental cost-effectiveness ratio for this is $174 per case detected and treated for pre-cancer.

While the clinical trial protocol calls for all returning patients with VIA-/HPV- results to have a colposcopy at the second visit, we also conducted a scenario analysis where fewer women receive colposcopy. If we assume that only women with high-risk HPV genotypes 16 or 18 and a positive or negative VIA result have a colposcopy done, this further reduces the detection cost in the intervention arm. The average cost per patient screened falls from $12.5 to $5.95, but only 140 additional pre-cancer or cervical cancer cases are detected and treated compared to control, rather than the additional 247 in the higher biopsy use scenario. The incremental cost-per-case detected and treated falls to $103, so, in the short-term, it would be cost-saving for the 140 patients, but cost-increasing for the health system for the 107 untreated patients.

In terms of a limited societal perspective (Table 6), if we assume the cost of the patient time at the clinic plus transportation costs of $6 per visit, the higher number of visits (4,394) in the control arm would imply an additional aggregate cost of $26,365. From a societal perspective, this would offset some of the additional $125,000 in healthcare costs incurred for patients using the mobile application.

**Table 4. Clinical consequences—Detection and treatment.**

|  | Control | Mobile App | Difference |
|---|---|---|---|
| **Total Receiving Pre-Cancer Treatment** | Cases | Cases | Cases |
| CIN2/CIN3 following Vl Biopsy | 64 | 72 | 8 |
| Colposcopy Abnormal-Pre-CA Tx | 346 | 518 | 173 |
| V2 Colposcopy Abnormal-CIN2/CIN3 | 32 | 78 | 45 |
| V2 Suspicious Ca—CIN2/3 | 1.8 | 4.9 | 3.1 |
| *Total Detected Cases with Pre-CA Tx* | ***444*** | ***673*** | ***229*** |
| **Total Receiving Cancer Treatment** |  |  |  |
| V2 Colposcopy Abnormal-Suspicious for CA | 3 | 7 | 5 |
| Screening visits (suspicious of CA-Cancer) | 68 | 77 | 9 |
| V2 Suspicious for Ca—Cancer | 3 | 7 | 5 |
| *Total Detected Cases with Cancer Tx* | ***73*** | ***91*** | ***18*** |
| **Cases Dx& Tx** | ***517*** | ***764*** | ***247*** |

Source: Authors' estimates.

**Table 5. Incremental cost-effectiveness analysis.**

| Trial period | Estimates |
|---|---|
| Total Detection and Treatment Costs |  |
| Total | $125,278 |
| *Incremental (Per Screened Member)* | *$12.53* |
| Incremental Cost of Screening and Detection |  |
| Total | $42,970 |
| *Incremental (Per Screened Member)* | *$4.30* |
| Incremental Cases Detected | 247 |
| *Incremental Cost Per Case Detected* | *$174* |

Source: Authors' estimates.

**Table 6. Costs-limited societal perspective.**

|  | Control arm | | Mobile app | | Difference |
|---|---|---|---|---|---|
|  | *Number* | *Total* | *Number* | *Total* |  |
|  |  | Cost |  | Cost |  |
| Total Direct Medical Costs | – | $651,454 | – | $776,732 | $125,278 |
| Indirect/Time Costs |  |  |  |  |  |
| First Visits | 10,000 |  | 10,000 |  |  |
| Second Visits | 5,880 |  | 1,620 |  |  |
| Third visits (incl Tx) | 296 |  | 162 |  |  |
| Total | 16,176 | $97,056 | 11,782 | $70,691 | $26,365 |
| Total Societal Costs |  | $748,510 |  | $763,143 | $14,633 |

Source: Authors' estimates.

## Discussion

This prospective clinico-economic model projects the potential impact of a mobile app for cervical cancer screening on health system costs and clinical outcomes and aims to inform policy and program decision-making in Kenya and other high-burden settings. The overall projection is that for a cohort of 10,000 women enrolled in an HPV-based cervical cancer screening program, an app-based intervention would incur $125,278 more ($12.53 per patient) during a three-visit screening process. The health consequences would be that an additional 247 women would be detected early and receive pre-cancer treatment and an additional 18 women would be detected and receive cancer treatment.

The cost-consequences assessment was able to model patient flow through a standard clinical workflow process in Kenya, accounting for screening visits, follow-up visits, and expected costs of care influenced by assumptions related to lost-to-follow-up rates. The use of a mobile phone app is estimated to result in fewer "unnecessary" post-screening follow-up visits due to the ability to inform patients about normal or negative test findings, complemented with educational messages and subsequent follow-up testing recommendations. Due to an assumed higher likelihood of app-using patients returning for cervical cancer-related second and third healthcare encounters, more patients who have confirmed HPV+ results in the mobile application arm were expected to receive guideline-recommended colposcopy, biopsy, and eventual pre-cancer or cancer treatment. Although there were differential impacts between the arms in the expected clinical workflow, there were also higher overall costs for using the mobile app. However, these higher costs are due to more patients receiving clinically appropriate care, guided by the ease of patient-level communication through the app. Although not modeled specifically, there are likely secondary benefits for a busy clinic in the reduction of congestion from women who can avoid returning to the clinic to learn that they have negative/normal cervical cancer screening results.

While this analysis was conducted primarily as a short-term cost-consequences projection, we were also able to calculate a cost-effectiveness ratio for diagnostic costs generated up to the point of pre-cancer treatment. The calculations imply that the incremental cost per patient detected and treated is $174, and $103 if only hrHPV (16/18) patients receive a colposcopy. Is this good value for money from a health system perspective? To answer this, it must be compared with the lifetime costs and survival for a woman who progresses to invasive cervical cancer. Our analysis did not estimate this nor could we find specific Kenyan estimates for these two outcomes although we were able to use more aggregated Kenyan cost data for cancer screening and treatment for comparison purposes and to support our parameter values and cost ranges. A study in Tanzania estimated per-patient lifetime discounted cervical cancer treatment costs ranging from $1,700 (for Stage 4) to $3,000 (for Stage 1) [28]. The study estimated that Stage 1 patients lost about 1.1 years of life and Stage 4 patients lost about 6 years of life. Several Kenyan cost studies suggest that these treatment cost estimates would be comparable in Kenya. Studies have often reported "patient out-of-pocket" costs as the source of payment for services in Kenya [29] which is different than estimating limited-societal costs related to "producing or delivering" a healthcare visit or care episode—the focus of our analysis. Studies assessing patient costs in several cancer types estimated that cervical cancer treatment costs vary and increase by cancer stage severity. Treatment costs were reported to range from approximately $1,000 to several thousands of dollars, with higher payments (costs) for care at private clinics compared to public facilities [23,30]. If these ranges of estimates were to apply in Kenya, the impact of the mobile app would be judged to be not only cost-saving but also "dominant" in economic terminology—i.e., lower cost and better outcomes. That would represent good value for money from a healthcare system perspective and is likely to be favorably viewed

by health ministries and funders. The basic intuition behind this extrapolation is that the mobile app arm generates $42,970 in additional aggregate costs during a shorter-term diagnostic phase to detect 247 additional women with cervical cancer, but, for example, with an average lifetime cost per case of $2,000, this would save almost $600,000 in cancer treatment costs even without accounting for the value of the additional life years gained.

Several studies have examined the economic aspects of incorporating digital tools into cervical cancer screening in Kenya [23,31]. The cost-related studies have mostly reported on "costs per woman screened" and other micro-costs associated with community-based and clinic-based screening strategies in Kenya [32,33]. Those studies focused on community health campaigns to improve screening and compared screening effects and costs for these short-term campaigns (30 days total for outreach, screening, and notification/referral) to outcomes in government health clinics, assessing HPV-based screening using self-collection strategies in these settings [23,31]. This study complements the previous work, focusing on a new intervention to improve screening efficacy and efficiency. Our analysis used data collected in public and private clinic-based settings to model a more comprehensive process of VIA and HPV-based testing, followed by subsequent diagnoses for persons based on their screening results (i.e., colposcopy and biopsy when appropriate), as well as delivery of pre-cancer and cancer treatments based on patient's health profiles through this guideline-based testing regimen. Our prospective model is consistent with recommended processes starting from screening and carrying through to treatment initiation for patients identified as having pre-cancerous lesions or diagnosed with cervical cancer.

Other cost-effectiveness studies have focused on expanding HPV-based cervical cancer screening and treatment strategies; however, study participants were often patients with ongoing care at an HIV-related clinic [32,34]. Our modeling study of the potential value provided by a cervical cancer screening phone-based app enabled the testing of various scenarios, including our base-case scenario and other select sensitivity analyses. We were able to utilize parameter estimates and cost ranges from the published screening-based and treatment literature, and to supplement these estimates with direct data collection of targeted time, resource needs, cost, and probabilities associated with our model structure of cervical-cancer patient flow and clinician workflow in Kenya.

This prospective economic modeling study has several limitations. First, as a preliminary analysis, it is hindered by the use of "best expert judgment" or "best literature-based estimate" informing parameter values in the model. The purpose of prospective modeling is to forecast scenarios and likely outcomes, but a drawback is that the model has limited study-based parameter values to use. A related limitation is that modelers must rely on literature-based assumptions for clinical and economic parameters, and there is generally substantial heterogeneity in the types of studies reported in medical literature. Studies vary in the types of patients assessed—e.g., HIV populations or specialty clinics in the case of cervical cancer—or geographic location of patients, and the types of interventions tested and/or compared in published studies. That said, there are several high-quality publications related to cervical cancer screening in Kenya, and we used available HPV-based test outcomes reported, clinical status classifications, follow-up rates, and cost estimates provided by the research community. Our model structure focused on a specific three-visit trial design using laboratory-based HPV-DNA testing which was consistent with guidelines-based practice in Kenya (i.e., screen, further diagnose if risk-signal present, treat when appropriate). In this short-term model, we did not model longer-term costs or effects related to multi-year visit schedules, and subsequent progression of cancer cases missed in diagnosis or not treated, nor did we estimate mortality reduction or lifetime morbidity effects for the mobile application compared to standard of care approaches. Nor did we model different regions in Kenya (urban vs. rural or western

Kenya vs. eastern Kenya) or assess potential cost-consequence differences in public vs. private clinics or via alternative screening campaign strategies.

A more complete economic assessment would include formal longer-term cost-effectiveness modeling and potential consideration of impacts on clinic operations and of regional or national scale-up of the intervention [35,36]. Further prospective population-level modeling should be considered for future studies to understand broader lifetime expected population health profiles, and potential impacts of digital health interventions to support cervical cancer screening programs on resource-constrained health systems.

Our model was aligned with an RCT study aiming to assess the clinical consequences associated with using a mobile app compared to standard of care to aid clinicians and patients in disseminating and receiving information, respectively, about cervical cancer screening outcomes related to HPV testing and follow-up care recommendations. Plausible parameter ranges informed this analysis, and it will be important to reanalyze the intervention with trial-based results for critical parameters. The most critical ones are those related to whether women receiving the app-based mobile intervention are more likely to return and receive appropriate follow-up care.

## Conclusion

This prospective clinico-economic model, constructed in conjunction with an ongoing HPV-based cervical cancer screening trial, aimed to assess the potential clinical and economic impact of an innovative mobile phone-based app to communicate outcomes and recommendations to patients in western and eastern Kenya. The analysis suggested that under plausible assumptions about patient behavior, disease epidemiology, treatment patterns, and costs from prior literature and expert elicitation, the proposed intervention would be cost-increasing in the short term but result in greater detection and treatment. Given the projected increase in detection and treatment, the mobile app intervention is expected—over an average woman's lifetime—to be cost-saving for the health system while significantly improving patients' life expectancy. Thus, a mobile-app-based intervention could be cost-effective and an economically "dominant" intervention(i.e., improving outcomes and reducing costs) and support achieving WHO cervical cancer disease elimination goals.

## Supporting information

**S1 Appendix. Screening visit and follow-up visit structure for prospective clinico-economic model.**
(DOCX)

**S1 File. Supporting inclusivity in global research questionnaire.**
(DOCX)

## Acknowledgments

We would like to thank and acknowledge the contributions of Kennedy Olweny and David Muyodi of the International Cancer Institute in Eldoret, Kenya for assistance with facilitating data collection, and Dr. Joseph Babigumira for helpful discussions related to the economic modeling approach and cancer screening processes in Africa.

## Author contributions

**Conceptualization:** Fredrick Chite Asirwa, Brian W. Bresnahan, Faith Yego, Dana Duncan, James K. Karichu, Louis P. Garrison Jr.

**Data curation:** Fredrick Chite Asirwa, Brian W. Bresnahan, Faith Yego, James K. Karichu, Louis P. Garrison Jr.

**Formal analysis:** Fredrick Chite Asirwa, Brian W. Bresnahan, Faith Yego, Dana Duncan, Louis P. Garrison Jr.

**Funding acquisition:** Louis P. Garrison Jr.

**Investigation:** Fredrick Chite Asirwa, Brian W. Bresnahan, Faith Yego, Dana Duncan, Louis P. Garrison Jr.

**Methodology:** Fredrick Chite Asirwa, Brian W. Bresnahan, Faith Yego, Dana Duncan, James K. Karichu, Louis P. Garrison Jr.

**Project administration:** Brian W. Bresnahan, Faith Yego, Louis P. Garrison Jr.

**Resources:** Fredrick Chite Asirwa, Brian W. Bresnahan, Faith Yego, Dana Duncan, James K. Karichu, Louis P. Garrison Jr.

**Software:** Brian W. Bresnahan, Faith Yego, Louis P. Garrison Jr.

**Supervision:** Louis P. Garrison Jr.

**Validation:** Fredrick Chite Asirwa, Brian W. Bresnahan, Faith Yego, Dana Duncan, James K. Karichu, Louis P. Garrison Jr.

**Visualization:** Brian W. Bresnahan, Faith Yego, Dana Duncan, James K. Karichu, Louis P. Garrison Jr.

**Writing – original draft:** Fredrick Chite Asirwa, Brian W. Bresnahan, Faith Yego, Dana Duncan, James K. Karichu, Louis P. Garrison Jr.

**Writing – review & editing:** Fredrick Chite Asirwa, Brian W. Bresnahan, Faith Yego, Dana Duncan, James K. Karichu, Louis P. Garrison Jr.

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
