## [Decision Letter · Decision Letter 0]

1 Oct 2024

PONE-D-24-32009A Prospective Model of the Potential Clinical and Economic Impact of

Cervical Cancer Screening Supported by a Mobile Phone AppPLOS ONE

Dear Dr. Garrison,

Thank you for submitting your manuscript to PLOS ONE. After careful consideration, we feel that it has merit but does not fully meet PLOS ONE’s publication criteria as it currently stands. Therefore, we invite you to submit a revised version of the manuscript that addresses the points raised during the review process.

We look forward to receiving your revised manuscript.

Kind regards,

Edward Adekola Oladele, MD, MPH, PhD

Academic Editor

PLOS ONE

Journal Requirements:

2. Please include a complete copy of PLOS’ questionnaire on inclusivity in global research in your revised manuscript. Our policy for research in this area aims to improve transparency in the reporting of research performed outside of researchers’ own country or community. The policy applies to researchers who have travelled to a different country to conduct research, research with Indigenous populations or their lands, and research on cultural artefacts. The questionnaire can also be requested at the journal’s discretion for any other submissions, even if these conditions are not met.  Please find more information on the policy and a link to download a blank copy of the questionnaire here: https://journals.plos.org/plosone/s/best-practices-in-research-reporting . Please upload a completed version of your questionnaire as Supporting Information when you resubmit your manuscript.

3. In the ethics statement in the Methods, you have specified that verbal consent was obtained. Please provide additional details regarding how this consent was documented and witnessed, and state whether this was approved by the IRB

4. Please note that PLOS ONE has specific guidelines on code sharing for submissions in which author-generated code underpins the findings in the manuscript. In these cases, we expect all author-generated code to be made available without restrictions upon publication of the work. Please review our guidelines at https://journals.plos.org/plosone/s/materials-and-software-sharing#loc-sharing-code and ensure that your code is shared in a way that follows best practice and facilitates reproducibility and reuse.

5. Thank you for stating the following financial disclosure: Roche Diagnostics funded the economic analysis through a contract with Veritech Corporation, providing support for LPG, BWB, and FY.   

Reviewers' comments:

Reviewer's Responses to Questions

**Comments to the Author**

1. Is the manuscript technically sound, and do the data support the conclusions?

Reviewer #1: Yes

Reviewer #2: Partly

2. Has the statistical analysis been performed appropriately and rigorously? 

Reviewer #1: N/A

Reviewer #2: Yes

3. Have the authors made all data underlying the findings in their manuscript fully available?

Reviewer #1: Yes

Reviewer #2: Yes

4. Is the manuscript presented in an intelligible fashion and written in standard English?

Reviewer #1: Yes

Reviewer #2: Yes

5. Review Comments to the Author

Reviewer #1: The paper does a good job explaining the limitations of this type of modeling tool while also highlighting the important role played by preliminary models like this one. You also highlighted some additional data that would be useful for future studies.

Reviewer #2: The authors have picked an interesting topic - the use of mobile technology in the followup of women being screened for Cervical Cancer, an important health problem. The study has been generally well conducted but I would like to address the following.

1. A clear definition of the time horizon over which costs and consequences were accrued is lacking, which is different from the study period.

2. A clear definition of the consequences being considered is not outlined. It is summarized as clinical, workflow and guideline pathways which is inadequate.

3. Also the phrase " standardized to 2022 cost" is not appropriate. This is because are cost are adjusted to a particular year to account for inflation and there are different approaches which can be used and the choice of approach should be specified.

4. Discounting is not mentioned and the rationale for not doing so should be provided.

6. PLOS authors have the option to publish the peer review history of their article (what does this mean? ). If published, this will include your full peer review and any attached files.

**Do you want your identity to be public for this peer review?** For information about this choice, including consent withdrawal, please see our Privacy Policy .

Reviewer #1: **Yes: ** Meghan Steel

Reviewer #2: No

---

## [Author Response · Author response to Decision Letter 0]

3 Dec 2024

Response to Reviewers

Reviewer's Responses to Questions and Our Responses to Reviewers.

Comments to the Author

1. Is the manuscript technically sound, and do the data support the conclusions?

Reviewer #1: Yes

Reviewer #2: Partly

Authors' response:

Thanks! We address the four specific questions raised by Reviewer #2 below.

2. Has the statistical analysis been performed appropriately and rigorously?

Reviewer #1: N/A

Reviewer #2: Yes

Authors' response:

Thanks!

3. Have the authors made all data underlying the findings in their manuscript fully available?

Reviewer #1: Yes

Reviewer #2: Yes

Authors' response:

Thanks! The data are provided in the manuscript.

4. Is the manuscript presented in an intelligible fashion and written in standard English?

Reviewer #1: Yes

Reviewer #2: Yes

Authors' response:

Thanks! We appreciate this. We have corrected some minor errors that are visible in the marked-up version.

5. Review Comments to the Author

Reviewer #1: The paper does a good job explaining the limitations of this type of modeling tool while also highlighting the important role played by preliminary models like this one. You also highlighted some additional data that would be useful for future studies.

Authors' response:

Thanks! We appreciate your understanding that this is a prospective clinico-economic evaluation using modeling.

Reviewer #2: The authors have picked an interesting topic - the use of mobile technology in the follow up of women being screened for Cervical Cancer, an important health problem.

The study has been generally well conducted but I would like to address the following.

Authors' response:

Thanks! We appreciate these questions:

1. A clear definition of the time horizon over which costs and consequences were accrued is lacking, which is different from the study period.

Authors' response:

Thanks! The analytic time horizon is the period (less than one year) for a typical patient to complete the three visits. It is shorter than the study period. From a practical perspective, our modeling approach assessed “events” occurring or not occurring (such as follow-up visits), rather than specific number of days or weeks/months. Timing of second or third visits, for example, varied. This is a realistic scenario, particularly in a low-to-middle-income country setting. We have added this language on page 5, line 98 to make it clear: "(usually completed in less that 6 months)"

2. A clear definition of the consequences being considered is not outlined. It is summarized as clinical, workflow and guideline pathways which is inadequate.

Authors' response:

Thanks for this comment. We have revised the text on page 6 line 104-5 to address this, adding: " "Key consequences include intermediate endpoints (such as test results) and final endpoints (viz., pre-cancerous and cancer cases detected and treated)." And on page 7, line 128-9, adding: " such as cost-per-case detected and cost-per-cancer-case avoided)'.

3. Also the phrase " standardized to 2022 cost" is not appropriate. This is because are cost are adjusted to a particular year to account for inflation and there are different approaches which can be used and the choice of approach should be specified.

Authors' response:

Thanks for this comment. We agree that the word "standardized' is unclear. We have changed it to "estimated costs in 2022," which more accurately reflects our approach.

Medical cost finding for economic evaluation from a (limited) societal perspective is difficult in both developed and developing counties if the aim is to use opportunity costs for clinical and economic policy making. Prices in medical "markets" are affected by government price setting and insurance--both public or private.

For this prospective modelling exercise, we needed to have plausible estimates or assumptions for the model parameters. By necessity, we used cost information from multiple sources to generate plausible base-case assumptions and ranges. Thus, our cost estimates were "triangulated" from multiple input sources, including: prospective data collection at sites, literature-based estimates, and expert opinion at the primary clinical site of the principal investigator in Kenya.

The prospective cost-related data was collected in 2022 in multiple Kenyan sites, including with hospital and clinic administrators (as IRB approved). Thus, we report our costs as 2022 U.S. dollars, converted from Kenyan shillings. We also used a time-and-motion estimation approach in 2022 to collect supporting data related to resources used during visits, time, and costs.

We did not apply a specific formulaic inflation adjustment to the literature-based costs estimates for visits, tests, or therapies, which were published in various years. We applied "face-validity testing" with experts (clinical and economic) to our 2022 base-case cost estimates and to our ranges of costs. We assessed how our estimates compared with recent published ranges for variables similar to our model inputs for amounts and ranges of cost.

A challenge is that literature-based costs varied in the inputs included in various studies, depending on the research question being asked and addressed, and other regional or national differences. Also, unlike the availability in the United States for having annual medical care price change percentages, there is not a similar precise "medical care inflation-adjustor." In Kenya, general inflation data is available through the Kenya National Bureau of Statistics (KNBS), and healthcare-related components might be included in the broader Consumer Price Index (CPI). However, these components typically lack the granularity or specificity of a dedicated "medical care inflation" index, making it more challenging to precisely adjust healthcare costs annually.

4. Discounting is not mentioned and the rationale for not doing so should be provided.

Authors' response:

Thanks for spotting this omission. We have added this sentence on page 9, lines 180-1, to make address this: "Since the time horizon for the primary analysis is short-term (i.e., less than one year to complete the 3 visits), there is no strong need to perform discounting of costs or outcomes."

Inquiry under Journal Requirements

3. In the ethics statement in the Methods, you have specified that verbal consent was obtained. Please provide additional details regarding how this consent was documented and witnessed, and state whether this was approved by the IRB.

Authors' Response:

We have added on page 9, lines 174-7: "For the verbal consent process, the study was explained to the participants who were selected based on their expertise in the study topic, and once they agreed to participate, the consent was documented and the interview conducted. All informed consenting processes were approved by the IRB."

---

## [Editor Report · Decision Letter 1]

5 Dec 2024

A Prospective Model of the Potential Clinical and Economic Impact of

Cervical Cancer Screening Supported by a Mobile Phone App

PONE-D-24-32009R1

Dear Dr. Garrison,

We’re pleased to inform you that your manuscript has been judged scientifically suitable for publication and will be formally accepted for publication once it meets all outstanding technical requirements.

Kind regards,

Edward Adekola Oladele, MD, MPH, PhD

Academic Editor

PLOS ONE

---

## [Editor Report · Acceptance letter]

PONE-D-24-32009R1

PLOS ONE

Dear Dr. Garrison Jr.,

I'm pleased to inform you that your manuscript has been deemed suitable for publication in PLOS ONE. Congratulations! Your manuscript is now being handed over to our production team.

Kind regards,

on behalf of

Dr. Edward Adekola Oladele

Academic Editor

PLOS ONE